A validation of Emotiv EPOC Flex saline for EEG and ERP research

http://orcid.org/0000-0003-4349-906X Williams Nikolas S. 1 nikolas.williams@mq.edu.au
http://orcid.org/0000-0003-1912-820X McArthur Genevieve M. 1
http://orcid.org/0000-0002-3731-1887 de Wit Bianca 1
Ibrahim George 1
http://orcid.org/0000-0001-6862-4694 Badcock Nicholas A. 1 2
1 Department of Cognitive Science, Macquarie University , Sydney, NSW , Australia
2 School of Psychological Science, University of Western Australia , Perth, WA , Australia
Gollo Leonardo
Electronic publication date: 2020 Aug 11
Publication date: 2020
Volume: 8
Electronic Location ID: e9713
Received 2020 Mar 2; Accepted 2020 Jul 23
Copyright: © 2020 Williams et al.
Copyright year: 2020
Copyright holder: Williams et al.
License: This is an open access article distributed under the terms of the Creative Commons Attribution License, which permits unrestricted use, distribution, reproduction and adaptation in any medium and for any purpose provided that it is properly attributed. For attribution, the original author(s), title, publication source (PeerJ) and either DOI or URL of the article must be cited.
License URL: https://creativecommons.org/licenses/by/4.0/

Keywords: EEG, ERP, Emotiv, Validation, MMN, P300, SSVEP, EPOC Flex, N170, Alpha

Funding: Neural Markers of Learning Success Industry 83673928 Macquarie University and Emotiv Pty Ltd This work was supported by the Neural Markers of Learning Success industry partnership grant (No. 83673928) between Macquarie University and Emotiv Pty Ltd. The funders had no role in study design, data collection and analysis, decision to publish, or preparation of the manuscript.

==============================
Background

Previous work has validated consumer-grade electroencephalography (EEG) systems for use in research. Systems in this class are cost-effective and easy to set up and can facilitate neuroscience outside of the laboratory. The aim of the current study was to determine if a new consumer-grade system, the Emotiv EPOC Saline Flex, was capable of capturing research-quality data.

Method

The Emotiv system was used simultaneously with a research-grade EEG system, Neuroscan Synamps2, to collect EEG data across 16 channels during five well-established paradigms: (1) a mismatch negativity (MMN) paradigm that involved a passive listening task in which rare deviant (1,500 Hz) tones were interspersed amongst frequent standard tones (1,000 Hz), with instructions to ignore the tones while watching a silent movie; (2) a P300 paradigm that involved an active listening task in which participants were asked to count rare deviant tones presented amongst frequent standard tones; (3) an N170 paradigm in which participants were shown images of faces and watches and asked to indicate whether the images were upright or inverted; (4) a steady-state visual evoked potential (SSVEP) paradigm in which participants passively viewed a flickering screen (15 Hz) for 2 min; and (5) a resting state paradigm in which participants sat quietly with their eyes open and then closed for 3 min each.

Results

The MMN components and P300 peaks were equivalent between the two systems (BF10 = 0.25 and BF10 = 0.26, respectively), with high intraclass correlations (ICCs) between the ERP waveforms (>0.81). Although the N170 peak values recorded by the two systems were different (BF10 = 35.88), ICCs demonstrated that the N170 ERP waveforms were strongly correlated over the right hemisphere (P8; 0.87–0.97), and moderately-to-strongly correlated over the left hemisphere (P7; 0.52–0.84). For the SSVEP, the signal-to-noise ratio (SNR) was larger for Neuroscan than Emotiv EPOC Flex (19.94 vs. 8.98, BF10 = 51,764), but SNR z-scores indicated a significant brain response at the stimulus frequency for both Neuroscan (z = 12.47) and Flex (z = 11.22). In the resting state task, both systems measured similar alpha power (BF10 = 0.28) and higher alpha power when the eyes were closed than open (BF10 = 32.27).

Conclusions

The saline version of the Emotiv EPOC Flex captures data similar to that of a research-grade EEG system. It can be used to measure reliable auditory and visual research-quality ERPs. In addition, it can index SSVEP signatures and is sensitive to changes in alpha oscillations.

Introduction

Brain activity creates changes in electrical potentials. The pattern of these changes, or oscillations, can be recorded from sensors placed on the scalp and results in an electroencephalogram (EEG; Nunez & Srinivasan, 2006). EEG is a powerful method for investigating cognitive phenomena as it can non-invasively index neural processes that are too fleeting to be measured using other brain-imaging techniques. EEG can be analysed according to how often oscillations occur or when oscillations occur. The first approach is considered spectrum analyses in which oscillations are characterised according to their frequencies, with specific frequency bands denoted by particular terms. For example the delta band is the slowest-wave frequency (less than 4 Hz) and is typically observed during deep sleep (Amzica & Steriade, 1998), whereas the alpha band (8–12 Hz) has been associated with myriad processes including attention (Ray & Cole, 1985; Sauseng et al., 2005; Uusberg et al., 2013), relaxation (Jacobs & Friedman, 2004), and listening effort (McMahon et al., 2016). Changes in EEG frequencies can be used to investigate clinical phenomena such as depression (Debener et al., 2000) or Alzheimer’s disease (see Vecchio et al. (2013), for a review), as well as cognitive processes such as memory and attention (Bonnefond & Jensen, 2012; Düzel, Penny & Burgess, 2010; Klimesch, 1999). Even simple resting state paradigms, in which frequency bands are compared between ‘eyes open’ and ‘eyes closed’ conditions, can answer questions concerning psychiatric disorders (see Newson & Thiagarajan (2019) for a review), autism (Billeci et al., 2013), and ageing (Barry & De Blasio, 2017).

In contrast to the frequency of brain waves, EEG research is often concerned with the timing of the brain’s response to specific stimuli. This can be represented by event-related potentials (ERPs) which are the average waveform of electrical activity that occurs immediately before and after a stimulus of interest. An auditory ERP is the average response to a particular sound (e.g. a tone) that is typically characterised by a waveform with three positive peaks, P100, P200, and P300, and two negative peaks, N100 and N200 (Luck, 2014). Differences in these waveforms between conditions or populations can indicate differences in auditory processing. For example changes in auditory stimuli, such as the onset of a high-pitched ‘deviant’ tone after a sequence of low-pitched ‘standard’ tones, will generally elicit a change in neural response—even if a participant is not actively attending to the tones. Subtracting the response to the deviant tone from the standard tone reveals the mismatch negativity (MMN) component, which reflects an automatic brain response to change detection in passive auditory processing (see Garrido et al. (2009), Näätänen et al. (2007) for reviews).

Event-related potentials can also be used to study higher-level cognitive processing. For example when a participant is asked to actively attend to a sequence of standard and deviant tones, the onset of the deviant tone will produce a particularly large third peak in the auditory ERP. This P300 (or P3) response is thought to be reflective of higher-level cognitive processing such as working memory (Luck, 1998; Vogel & Luck, 2002; Vogel, Luck & Shapiro, 1998) and attention (see Polich (2007), Polich & Kok (1995) for reviews). The robust and distinctive nature of a P300 makes it a useful signature for investigating cognitive processing in target-detection tasks.

Event-related potentials can also be used to investigate cognitive processing in the visual domain. For example the face-sensitive N170 occurs when a participant is visually-presented with an object. Participants will typically exhibit an ERP waveform with a negative deflection occurring approximately 170 ms after the presentation. This deflection is generally larger for faces than other objects (Eimer, 2011). In addition, the N170 is also sensitive to structural changes in faces, such as a delay in onset when faces are inverted (Itier, Latinus & Taylor, 2006; Itier & Taylor, 2004; Rossion et al., 2000) and an enhanced amplitude when faces show emotion (see Hinojosa, Mercado & Carretié (2015) for a meta-analysis).

Related to both frequency-based EEG studies and ERPs are steady-state visual evoked potentials (SSVEPs). Like ERPs, SSVEPs are produced as a result of external stimuli. Unlike ERPs, they are analysed on the frequency, rather than time, domain (see Norcia et al. (2015) for a review) and represent a phasic response to periodic stimulus, such as a flickering screen. SSVEPs will generally produce a large EEG signature at the same frequency as the stimulus, making them useful for investigating cognitive processes such as reading (Lochy, Van Belle & Rossion, 2015), face perception (Liu-Shuang, Norcia & Rossion, 2014), or to characterise mental state or intent in brain-computer interfaces (Alamdari et al., 2016). For example Liu-Shuang, Norcia & Rossion (2014) presented participants with a rapid series of faces in which every fifth face was different from the others. Not only did they report typical SSVEPs at the stimulus frequency, but also a larger SSVEP at the odd-face frequency, suggesting participants’ brains could discriminate the odd face even if the participants were unaware that the faces were different.

As they covertly measure neural processes, EEG and ERP studies are particularly useful in populations for whom laboratory conditions may be inappropriate or untenable. For example both auditory and visual ERPs may be used to investigate cognitive processing in schizophrenia (Feuerriegel et al., 2015), dyslexia (McArthur, Atkinson & Ellis, 2009; Peter et al., 2019), autism (Kim et al., 2018; Schwartz, Shinn-Cunningham & Tager-Flusberg, 2018) or neurodegeneration (Morrison et al., 2019; Swords et al., 2018). This highlights three drawbacks of traditional EEG systems (e.g. Neuroscan): (1) the involved setup procedures, which often include hair-washing and brushing, scalp abrasion, and the application of messy gels; (2) the limitation to lab-based research, as many research-grade systems are composed of large headboxes and amplifiers and transmit data via physical cables; and (3) cost, as research-grade systems often cost tens of thousands of dollars, putting them out of reach for many under-funded researchers. Consumer-grade EEG systems offer a potential solution to these issues. Several companies have released relatively inexpensive EEG systems aimed at the gaming and neurofeedback market that are typically portable and much simpler to set up than research-grade systems. For example Emotiv EPOC®, Muse2, and NeuroSky MindWave all cost less than $700 USD, are wireless, and use either dry electrodes or saline-soaked felt pads as the conduction medium. This contrasts with research-grade systems that cost much more and often use metal sensors that require the application of electro-conductive gel. In addition, research-grade systems often require heavy amplifiers that limit their portability.

Over the last decade, consumer EEG systems have attracted the attention of researchers. For example a recent scoping review of Emotiv EPOC applications found 382 journal articles and conference proceedings published from 2009 to 2019, including 51 experimental research studies and 31 validation studies (Williams, McArthur & Badcock, 2020). Debener et al. (2012) were among the first to investigate the utility of EPOC when they removed the amplifier from its original case and attached it to research-quality electrodes that were inserted into an EasyCap® (Herrsching, Germany). They then collected data from participants in both lab and outdoor (walking through a university campus) settings, with results demonstrating that the modified setup measured quality auditory ERPs. In 2014, some of these same researchers compared their modified EPOC setup to a research-grade system and found that P300 signatures were similar, suggesting that EPOC could capitalise on P300s for brain-computer interface applications (De Vos et al., 2014).

Other validations of the production-version EPOC were also published, with Duvinage et al. (2013) among the first to provide evidence that it could capture auditory P300 ERPs. In the same year, Badcock et al. (2013) reported that EPOC could be used to measure research-quality auditory ERPs in adults. Two years later, they replicated this in children (Badcock et al., 2015) and also validated EPOC to measure the face-sensitive N170 (De Lissa et al., 2015). In 2017, Barham et al. (2017) also validated EPOC for auditory ERPs in adults and Melnik et al. (2017) expanded that use to SSVEPs.

Other consumer-grade EEG devices have also been validated for experimental research. For example Krigolson et al. (2017) demonstrated that a Muse EEG system could be used to measure N200 and P300 ERPs in a visual oddball task and Ratti et al. (2017) showed that Muse and Mindwave could collect satisfactory resting state data in a clinical setting.

Although easy-to-setup consumer-grade systems can collect research-quality data, one of their biggest shortcomings is limited sensor density. For example Neurosky (1 channel), Muse2 (four channels), and EPOC (14 channels) have fewer sensors than most research-grade EEG systems (32, 64, or 128 channels). Further, the sensors of EPOC+, Muse2 and Mindwave are fixed in place (though see Debener et al. (2012) and De Vos et al. (2014) wherein researchers modified an EPOC to be configurable). Both the limited sensor count and rigid nature of the array limits the use of these devices to certain EEG or ERP paradigms. For example auditory ERPs, such as MMN and P300, are often largest over central sites (Fz and Pz; Garrido et al., 2009; Näätänen et al., 2007). Data collection at these sites is not possible with EPOC, Neurosky, or Muse2.

Recently, Emotiv released a new product, the Emotiv EPOC Flex (hereafter referred to as Flex), that provides greater sensor coverage (up to 32 channels) compared to other Emotiv products. In addition, the channel array is user-configurable according to the 10–20 international system, allowing researchers to optimise their EEG data collection for some paradigms. Another benefit of Flex is 14-bit resolution which, theoretically, provides greater signal-to-noise ratios compared to lower resolution devices such as Muse 2 (10-bit) and Mindwave (12-bit). These benefits of Flex are achieved while still maintaining the low cost and portability for which this device class is known. The saline sensor version of Flex also eliminates the need for electroconductive gel, which may not be desirable for certain populations or in settings outside of the laboratory. See Table 1 for a comparison of specifications of EEG devices.

Table 1 Comparison of technical specifications of EEG systems.

	Neuroscan SynAmps	EPOC+	EPOC Flex Saline	Muse 2	Mindwave	
Number of channels	64	14	32	4	1	
EEG Electrode locations	User configurable. International 10–20 system	Rigid. AF3, F3, F7, FC5, T7, P7, O1, O2, P8, T8, FC6, F8, F4, and AF4	User configurable. International 10–20 system	Rigid. AF7, TP9, TP10, and AF8	Rigid. FP1	
Reference locations	User configurable (online reference and ground)	TP9 and TP10 (CMS and DRL)	User configurable (CMS and DRL)	FPz (CMS and DRL)	Earclip (online reference and ground)	
Electrode material	Ag/AgCl	Saline-soaked felt pad	Saline-soaked felt pad	Silver (frontal) and silcone rubber (temporal)	Silver	
Resolution	24 bits	14 or 16 bits	14 bits	10 bits	12 bits	
Sampling rate	Up to 20 kHz	128 or 256 Hz	128 Hz	220 or 500 Hz	512 Hz	

Although Flex may fill a gap between expense, convenience, configurability, and sensor coverage, its capabilities as a research system are yet to be validated against a research-grade system. In this study, we compared Flex to a research-grade EEG system, Neuroscan SynAmps2, using a simultaneous setup across five research paradigms: the auditory MMN, the auditory P300, the face-sensitive N170, the SSVEP, and a resting state task.

Methods

The methods used to test participants were approved by the Macquarie University Human Research Ethics Committee (Ref: 5201831203493).

Participants

We used a Bayesian stopping rule to determine participant numbers. Our pre-registered plan (https://osf.io/b764p) was to calculate Bayes factors for a critical comparison for each task after we had tested 20 participants, and then for each participant thereafter, stopping each task when the Bayes factor was less than 0.33 (substantial evidence that the systems were equivalent; Jarosz & Wiley, 2014) or exceeded 3.00 (substantial evidence that the systems were different; Jarosz & Wiley, 2014).

For the auditory ERP tasks, we tested 17 females and 3 males from 18 to 52 years of age (M = 21.4; SD = 7.7) using the MMN paradigm, and 18 females and 2 males from 18 to 26 years of age (M = 19.25; SD = 1.9) for the P300 paradigm. For the N170, we tested 17 females and 3 males from 18 to 52 years of age (M = 21.4; SD = 7.7), and for the SSVEP paradigm, we tested 18 females and 2 males from 18 to 52 years of age (M = 20.9; SD = 7.6). For the resting state paradigm, we tested 18 females and 2 males from 18 to 52 years of age (M = 21.3; SD = 7.7). The participants characteristics differed between some tasks because we excluded participants from some tasks but not others. Excluded participants were replaced with new participants. See Table 2 for excluded participants and reasons for exclusions. All participants gave written informed consent to be involved in the research.

Table 2 Number of excluded participants for each paradigm with reasons for exclusion.

If a participant was excluded from one EEG system (e.g. Neuroscan), then they were also excluded them from the other (e.g. Flex).

Paradigm	Number excluded	Reason	
MMN	1	Extra events in Neuroscan EEG file (n = 1)	
P300	2	Greater than 50% rejected epochs (n = 1)
Missing events in Emotiv EEG file (n = 1)	
N170	2	Greater than 50% rejected epochs (n = 1)
Stimulus computer crash during task (n = 1)	
SSVEP	1	Experimenter error with recording EEG (n = 1)	
Resting	2	Missing events in Emotiv EEG file (n = 1)
Extra events in Neuroscan EEG file (n = 1)	

Stimuli

Stimuli and instructions were presented using Psychtoolbox (version 3.0.14; Brainard, 1997; Kleiner et al., 2007; Pelli, 1997) with MATLAB (version R2017b).

MMN paradigm

The stimuli—666 tones—were delivered using Phillips SHS4700/37 ear-clip headphones. Eight-five percent (566) were standard tones (1,000 Hz; 175 ms duration; 15 ms rise and fall time) and 15% (100) were deviant tones (1,500 Hz; 175 ms duration; 15 ms rise and fall time). To ensure that deviant tones were distributed throughout the experiment, 10 blocks were created. An equal number of deviant tones (i.e. 10) were assigned to each block and then tones were presented pseudo-randomly such that each block began with at least three successive standard tones and there were at least five (and no more than 35) standard tones between each of the deviant tones. Blocks were presented consecutively with no breaks between them. The stimulus-onset asynchrony (SOA) was randomly jittered between 900 and 1,100 ms to minimise stimulus anticipation effects. Participants were instructed to ignore the tones and watch a silent movie on an iPad. Thus, the MMN paradigm was considered a passive listening task and lasted approximately 11 min (666 s).

P300 paradigm

This paradigm was the same as the MMN paradigm except that participants were instructed to count and report the number higher-pitched (i.e. deviant) tones at the end of the session. Thus, the P300 paradigm was considered an active listening task.

N170 paradigm

Visual stimuli were viewed on an AOC 27-inch LCD monitor with a 60 Hz refresh rate at a distance of 50 cm from the participant. Stimuli were the same as a previous N170 validation study (De Lissa et al., 2015), which consisted of 75 unique face images (37 female faces, 38 male faces) and 75 unique watch images. Images were cropped using a standard-sized oval such that only internal face parts were visible. Each image was presented in both an upright and an inverted orientation condition. Thus, there were a total of 300 images presented across four conditions (upright faces, inverted faces, upright watches, inverted watches).

Each of 300 trials began with a white fixation cross presented for 500 ms against a dark background in the centre of the screen. This was followed by presentation of the face or watch stimulus for 200 ms. A blank screen was then presented until the participant indicated whether the image was upright or inverted (using the computer keyboard left and right arrows). After a response, a new screen instructed participants that they could blink their eyes. This blink screen lasted 1,500 ms and was followed by a 500 ms blank screen before the next trial began. The order of stimulus presentation was randomised at the outset of each experimental session and participants completed two blocks of 150 trials with a self-timed break between blocks.

SSVEP paradigm

The stimulus was a flickering screen that oscillated between grey (RGB values 160, 160, 160) and black (RGB values 0, 0, 0). The flickering screen was presented on the same monitor and with the same participant setup as the N170 paradigm. Our goal was to create a stimulus frequency near 15 Hz. Interactions between hardware and software meant that the measured frequency was 14.36 Hz. Participants watched the flickering screen for 2 min.

Resting state paradigm

In the eyes open condition participants were instructed to sit quietly and gaze at a black fixation cross against a grey (RGB values 160, 160, 160) background. In the eyes closed condition participants were instructed to sit quietly with their eyes closed until they heard a tone (which indicated the end of the trial). Each trial conditions lasted 3 min.

Neuroscan EEG system

We used 18 Ag–AgCl electrodes for the Neuroscan setup (SynAmps RT, Compumedics). These matched the Flex array and were placed in custom-cut holes immediately adjacent to each of the corresponding Flex electrodes along an axis toward Cz (for photograph of setup, see Supplemental Materials). Thus, the electrodes were located at Fp1, F3, FT7, CP3, P7, O1, O2, P8, CP4, FT8, F4, Fp2, Fz, Cz, Pz, Oz, left earlobe (online reference), and right earlobe (offline reference; see Fig. 1). Vertical electrooculogram (VEOG) electrodes were placed above and below the left eye and horizontal electrooculogram (HEOG) electrodes were placed at the outer canthus of each eye and the ground electrode was positioned between FPz and Fz. The Neuroscan data were sampled at 1,000 Hz and recorded to Curry software (version 7.0.10). Event-marking was achieved with parallel port triggers generated by the MATLAB presentation script using the io64 plugin (Scheiber, 2018).

Figure 1 Flex electrode positions for the validation study.

Letters represent Flex sensor labels. Blue/red font represents left/right side sensors.

Emotiv EPOC Flex EEG system

The Flex EEG electrode array consists of 32 Ag–AgCl electrodes, with 16 electrodes terminating in a ‘left’ wiring harness and 16 electrodes terminating in a ‘right’ wiring harness. In addition, there is a common-mode sensor (CMS; left side) and driven-right-leg (DRL; right side) sensor. The CMS and DRL serve as the online reference in ‘active’ sensor EEG setups like Flex. All electrodes accept saline-soaked felt pads and are attached to an EasyCap® (Herrsching, Germany) which is configurable according to the international 10–20 system.

To facilitate ease of set-up and clean-up, we modified the Flex system by removing eight unused electrodes from each side. The left-side sensors (and their positions) that we used were: A (Fp1), B (FT7), C (P7), D (O1), E (F3), F (CP3), G (Pz), H (Oz), and CMS (left earlobe). The right-side sensors (and their positions) that we used were: A (Fp2), B (FT8), C (P8), D (O2), E (F4), F (CP4), G (Fz), H (Cz), and DRL (right earlobe). See Fig. 1 for electrode locations.

Flex has in-built EEG data pre-processing including a high-pass filter of 0.2 Hz and a low-pass filter of 45 Hz, digitisation at 1,024 Hz and filtering using a digital 5th-order sinc filter, and downsampling to 128 Hz. EEG data were collected using Emotiv Pro (version 1.8.1).

Event-marking

One limitation of Flex is that it is not capable of native hardware event-marking. To time-lock stimuli to EEG data, we used another Emotiv product, Extender (www.emotiv.com/extender/). Extender is a hardware unit which has multiple functions: (1) It can be provide auxiliary power for Emotiv products; (2) It can record EEG data directly to a steady-state storage card; and (3) It can receive event-marking triggers via a 2.5 mm tip-ring sleeve audio jack. Using function 3, we delivered triggers to Extender via a BNC adaptor and coaxial cable. Triggers were then incorporated into the EEG data stream (see Fig. 2).

Figure 2 Flex validation setup schematic.

EEG data were collected simultaneously and acquired with Curry (Neuroscan) and Emotiv Pro (Flex) software. Event-marking was achieved using parallel port triggers.

Procedure

After cleaning the relevant areas on the face and earlobes, Neuroscan HEOG, VEOG, online reference (left earlobe), and offline reference (right earlobe) electrodes were attached to the participants. Electro-conductive gel was then inserted into each electrode. Next participants’ scalp was firmly combed to reduce electrode impedances (Mahajan & McArthur, 2010). Then the EasyCap, with all Neuroscan and Flex electrodes pre-fitted was placed on the participant. The experimenter then applied gel to each of the Neuroscan electrodes, taking care to first gently rub the tip of the blunt metal applicator syringe on the skin of the application site 3–4 times. Once all Neuroscan electrodes had been filled with gel, the experimenter applied 2–3 mL of saline solution to each of the Flex electrodes. Reference electrodes were attached to both the front (Flex; CMS and DRL) and back (Neuroscan; online reference and offline reference) of the earlobes. When all electrodes had been treated with a conductive agent, the impedances were measured. Neuroscan impedances were measured using the Curry software and adjustments to the gel were made until each electrode exhibited less than 5 kΩ. Using the Emotiv Pro software visualisation, we adjusted Flex electrodes until they were ‘green’, which indicates impedance values below 20 kΩ (G. Mackellar, 2020, personal communication).

Offline EEG general processing

All EEG data were processed using MATLAB and EEGLAB version 2019.0 (Delorme & Makeig, 2004). Spectrum analysis on SSVEP and resting state data was performed using MATLAB and Fieldtrip (version 20190819; Oostenveld et al., 2011). All processing code is available at Open Science Framework (https://osf.io/zj3f5/).

All EEG data were first bandpass filtered from 0.1 to 30 Hz. Neuroscan data were then downsampled to 129 Hz to match Flex data (see below regarding Flex sampling rate). Data were then re-referenced to the common average before continuing with paradigm-specific processing.

Flex sampling rate

Although the advertised sampling rate of Emotiv EPOC Flex is 128 Hz, practical use has suggested that sampling rates can vary from device to device. As variability in sampling rates can impact the ability to accurately measure EEG data, we measured the actual sampling rate of our particular Flex device. We did this by recording from the device for approximately 15 min. Using a custom-written script in MATLAB (available at Open Science Framework; https://osf.io/zj3f5), we then checked for any dropped samples (there were none) and calculated sampling rate as the number of samples collected divided by the total elapsed time in seconds. This resulted in a sampling rate of 129.05 Hz. We used this figure for all further calculations and analyses.

MMN processing

Following filtering, downsampling, and re-referencing to the common average, we removed eye-blink artefacts using independent component analysis (ICA) in EEGLAB (‘eeg_runica’ function). The pruned data were then epoched from −100 ms to 700 ms relative to stimulus onset. Each epoch was baseline corrected from −100 ms to 0 ms. Epochs with amplitude values ±150 μV were excluded. For consistency between systems, any epoch that was removed in one system was also removed in the other. Thus, only those epochs accepted in both systems were analysed. We then averaged accepted epochs to create ERP waveforms for both standard and deviant tones. We calculated each participants’ MMN by subtracting their standard waveform from their deviant waveform (Luck, 2014). The MMN peak for each EEG system was calculated as the average waveform value over the pre-determined time window (https://osf.io/b764p) of 100–200 ms following stimulus onset.

P300 processing

P300 EEG data were processed identically to MMN data with respect to ICA artefact removal, epoching and baseline removal, epoch rejection, and the creation of ERP waveforms for standard and deviant tones. However, the P300 waveform was the response to deviant tones only (i.e. we did not derive a component by subtracting waveforms as in the MMN). We calculated the P300 peak for each EEG system as the average waveform value over the pre-determined time window (https://osf.io/b764p) of 280–380 ms following stimulus onset.

N170 processing

EEG data in the N170 paradigm were processed similarly to MMN and P300 data with respect to ICA artefact removal, epoching and baseline removal, and epoch rejection. For each participant we created four waveforms: (a) Upright faces; (b) Inverted faces; (c) Upright watches; and (d) Inverted watches. The N170 peak was calculated as the average waveform value over the pre-determined time window (https://osf.io/b764p) of 120–220 ms.

SSVEP processing

After filtering, downsampling, and re-referencing we performed a spectral analysis with Fieldtrip using a single-taper fast Fourier transform and a 5 s, non-overlapping Hanning window in 0.2 Hz steps (‘ft_freqanalysis’ function with length = ‘5’, method = ‘mtmfft’, and taper = ‘hanning’ arguments). Using frequency-domain data, we calculated signal-to-noise ratio (SNR) values at each frequency bin. SNR was calculated as the ratio of the amplitude at each frequency to the average of the 20 surrounding bins (10 on each side, excluding the immediately adjacent bin; as performed in Liu-Shuang, Norcia & Rossion, 2014). We also calculated group z-scores for SNR values along the frequency spectrum (Liu-Shuang, Norcia & Rossion, 2014; Rossion et al., 2012).

Resting state processing

After filtering, downsampling, and re-referencing, we trimmed the data into eyes open and eyes closed trials. We then performed two spectral analyses for each participant on each trial type and for each system using a 5 s, non-overlapping Hanning window in 0.2 Hz steps in Fieldtrip (‘ft_freqanalysis’ function with length = ‘5’, method = ‘mtmfft’, and taper = ‘hanning’ arguments). One analysis was conducted on frequencies from 1 to 30 Hz in 1 Hz bins. The other analysis was conducted on alpha band (8–12 Hz) frequencies only, in 0.004 Hz bins. Group grand average power was calculated using ‘ft_freqgrandaverage’ and alpha band power scalp topography maps were created with Fieldtrip using ‘ft_topoplotER’.

Analysis

We exported all EEG data and performed analysis and visualisation in R (R Development Core Team, 2013) using tidyverse (Wickham et al., 2019) and BayesFactor (version 0.9.12-4.2; Morey & Rouder, 2018) packages. To test our main question of whether the two systems captured equivalent ERPs, we conducted Bayesian t-tests on specific channels for each task. We did this because we employed a Bayesian stopping rule and required a single statistical test for each task.

MMN analyses

We conducted a Bayesian t-test on the MMN components measured by each system at Fz. We chose Fz for two reasons: (1) The MMN is most robust in fronto-central regions (Garrido et al., 2009; Näätänen et al., 2007); and (2) We wished to test the unique capability of Flex, relative to other Emotiv EEG systems, for acquiring measurements along central locations.

To characterise the degree of similarity in waveforms captured by the two systems, we calculated intra-class correlations (ICC) between waveforms (Badcock et al., 2015; Badcock et al., 2013; Cassidy, Robertson & O’Connell, 2012; De Lissa et al., 2015; McArthur, Atkinson & Ellis, 2009; McArthur, Atkinson & Ellis, 2010). Between-system ICCs were calculated for both standard and deviant tone waveforms as well as for MMN waveforms. ICCs were deemed significant if the 95% confidence intervals did not include zero.

We also calculated measures of signal quality. The first of these measures was the number of rejected epochs (i.e. number of epochs with amplitude values ±150 μV) where a higher number of rejected epochs indicated poorer signal quality. We also calculated the signal-to-noise ratio (SNR) averaged across all 16 electrodes of each system. SNR was calculated as the root mean square of the signal (average amplitude between 100 and 200 ms) divided by the standard deviation of the baseline (−100 to 0 ms; Maidhof et al., 2009; Marco-Pallares et al., 2011). To compare the signal qualities between systems, we conducted a Bayesian t-test on the SNR values.

P300 analyses

We conducted a Bayesian t-test on the P300 peaks measured by each system at Pz. We chose Pz for two reasons: (1) The P300 ERP is most robust in parietal regions (Luck, 2014); and (2) We wished to test the unique capability of Flex, relative to other Emotiv EEG systems, for acquiring measurements along central locations. As with the MMN, we also calculated ICCs, the number of rejected epochs, and SNRs. We calculated P300 SNRs as the root mean square of the signal (average amplitude between 280 and 380 ms) divided by the standard deviation of the baseline (−100 to 0 ms; Maidhof et al., 2009; Marco-Pallares et al., 2011).

N170 analyses

We conducted a Bayesian t-test on the N170 peak at P8 in the upright faces condition. We chose P8 because the N170 is most robust over occipitotemporal regions, particularly over the right hemisphere (Eimer, 2011; Luck, 2014). Additionally, we calculated values for P7 and conducted a Bayesian t-test, though this analysis did not influence our pre-registered Bayesian stopping rule (https://osf.io/b764p). As with the MMN and P300, we also calculated ICCs, the number of rejected epochs, and SNRs. SNRs were calculated as the root mean square of the signal (120–220 ms) divided by the standard deviation of the baseline (−100 to 0 ms; Maidhof et al., 2009; Marco-Pallares et al., 2011).

We also compared the timing of the N170 peak for each participant by indexing the latency at which the minimum amplitude occurred during the time window of interest (i.e. 120 ms to 220 ms). This allowed us to compare not only latency of the N170 response between the systems, but also between stimulus conditions (i.e. face vs. watch and upright vs. inverted) as N170 latency is expected to be slightly, though robustly, delayed when faces are inverted relative to when they are upright (Eimer, 2000).

SSVEP analyses

We conducted a Bayesian t-test on the SNR, averaged across occipital (O1, Oz, O2) electrodes, at the stimulus frequency. We averaged the occipital electrodes because SSVEP generally originates from the visual cortex and is thus most pronounced over medial occipital areas (Norcia et al., 2015). We also calculated z-scores on the SNR spectra for each system at each frequency bin. To determine significance of SNR at each bin, we placed a z-score threshold at 1.96 (p < 0.05; Norcia et al., 2015).

Resting state analyses

We compared alpha power between the two systems by averaging alpha across occipital sites (O1, Oz, O2). We then performed a Bayesian ANOVA on alpha power with system (Neuroscan vs. Flex) and condition (eyes open vs. eyes closed) as factors.

Results

MMN results

Signal quality

For the MMN, one participant exceeded our threshold of 50% rejected epochs (333) and was removed from further analysis and replaced with a new participant. Table 3 displays descriptive statistics for epochs removed and SNRs. The Bayesian t-test between MMN SNRs suggested insufficient evidence to determine whether the systems differed. The number of rejected epochs was positively skewed so we used Wilcoxon Signed Rank Tests to compare the data between the two systems. Overall, there was a statistically greater number of Flex epochs rejected. Generally, the numbers of rejected epochs were low and there were sufficient data with which to conduct analyses.

Table 3 ERP signal quality of Neuroscan and Flex systems.

Median (inter-quartile range) [range] for number of rejected epochs and mean (standard error) [95% confidence interval] for signal-to-noise ratios (SNR). Epochs were rejected if amplitude exceeded ±150 μV.

		Epochs removed	SNR	
ERP	System	Median (IQR) [range]	Z	Mean (SE) [95% CI]	BF	
MMN	Neuroscan	0 (0) [0–47]	3.7*	4.15 (0.41) [3.35–4.95]	1.28	
	Flex	7.5 (33.8) [0–201]	3.54 (0.26) [3.04–4.05]	
P300	Neuroscan	0 (0) [0–53]	2.8*	6.60 (0.59) [5.43–7.76]	50.79	
	Flex	4.5 (20.5) [0–63]	5.10 (0.38) [4.37–5.84]	
N170	Neuroscan	0 (0) [0–24]	1.6┴	6.04 (0.47) [5.11–6.97]	7.84	
	Flex	0 (6.0) [0–23]	5.12 (0.29) [4.55–5.70]	
Notes:

* p < 0.01.

⊥ p = 0.11.

Waveform similarity

Figure 3 depicts a comparison between the grand average waveforms of each system according to listening condition (passive vs. active) and tone type (standard vs. deviant). ICC values can be found in their respective panels of Fig. 3. The waveforms used to create the MMN components (passive listening; A and C) were all very similar, with the ICCs greater than 0.80. We also calculated ICCs on the MMN component waveforms (Fig. 4). The waveforms were similar with an ICC of 0.82 and 95% confidence interval of 0.80 to 0.84. As none of the 95% confidence intervals included zero, we considered all ICCs to be significant.

Figure 3 Group auditory grand average ERP waveforms and intraclass correlations [95% confidence interval] for Neuroscan and Flex systems by listening condition and tone type.

Passive waveforms (A, C, E, and G) were used to create the MMN waveform. Deviant waveforms in the active listening condition (D and H) were the P300 waveforms. (A–D) depict channel Fz. (E–H) depict channel Pz. Intraclass correlations were considered significant if the 95% confidence interval did not include zero.

Figure 4 MMN component waveforms (deviant tones minus standard tones) for each system measured at Fz.

ICCs between the two waveforms were considered significant if the 95% confidence interval did not include zero. The shaded area denotes the pre-registered time window of interest.

MMN component peak

Figure 4 contrasts the MMN component waveforms between the two systems at Fz. We conducted a Bayesian t-test between the MMN magnitudes of each system (MMN magnitude was calculated as the average signal between 100 and 200 ms) to determine whether there was a difference between the components. Results favoured the null hypothesis with substantial evidence suggesting no difference between the MMNs captured by the two systems, BF10 = 0.25.

We also conducted a post-hoc analysis using a new time window that better represented the morphology of the MMN components observed in this study. To calculate the time window of measurement, we first calculated the time point of the minimum value of the overall grand average MMN waveform between 0 and 200 ms. We then centred a 100 ms interval on this figure and calculated MMN values. The time point of the minimum of the overall MMN waveform was 116 ms and thus the new time window of interest was 66 to 166 ms.

After calculating MMN peak values for each participant, we conducted Bayesian t-test between the systems. These results provided substantial evidence that there was no difference between the MMN peaks measured by the two systems, BF10 = 0.23. See Table 4 for descriptive statistics of MMN peaks using both the pre-registered and post-hoc calculations.

Table 4 Mismatch negativity (MMN) and P300 peak descriptive statistics for Neuroscan and Flex systems.

The MMN waveform was measured at Fz and calculated as the ERPs to deviant tones minus ERPs to standard tones in the passive condition. The pre-registered calculation represents the average signal from 100 to 200 ms whereas the post-hoc calculation represents the average signal from 66 to 166 ms. P300 values were measured at Pz and represent the average signal from either 280–380 ms (pre-registered) or 198–298 ms (post-hoc) for deviant tones in the active condition.

Calculation method	Waveform	System	Mean	SE	95% CI	
Pre-registered	MMN	Neuroscan	−1.03	0.30	[−1.61 to −0.45]	
	Flex	−0.97	0.38	[−1.71 to −0.22]	
	P300	Neuroscan	3.63	0.48	[2.67–4.58]	
	Flex	3.44	0.58	[2.30–4.58]	
Post-hoc	MMN	Neuroscan	−1.40	0.25	[−1.88 to −0.92]	
	Flex	−1.40	0.35	[−2.07 to −0.72]	
	P300	Neuroscan	3.22	0.43	[2.38–4.07]	
	Flex	3.56	0.51	[2.57–4.55]	

Table 5 shows the SNR of the MMN measured at Fz for each system. To calculate the signal, both the pre-registered and post-hoc methods were used. Figure 5 shows the MMN topographic distributions for Neuroscan (A) and Flex (B). Topographic values were calculated using the post-hoc time window only. A Pearson correlation suggested a moderate correlation of channel amplitudes between systems (r = 0.61, p < 0.001).

Figure 5 Topographic distribution of MMN and P300 ERP signals.

(A and B) Neuroscan and Flex MMN measured as the average signal between 66 and 166 ms. (C and D) Neuroscan and Flex P300 measured as the average signal between 198 and 298 ms.

Table 5 ERP signal-to-noise-ratios (SNRs).

Peak calculation method	ERP	Site	System	SNR (SE) [95% CI]	
Pre-registered	MMN	Fz	Neuroscan	5.51 (0.96) [3.64–7.39]	
			Flex	4.76 (0.71) [3.37–6.15]	
	P300	Pz	Neuroscan	11.3 (2.01) [7.39–15.3]	
			Flex	5.65 (0.92) [3.85–7.44]	
Post-hoc	MMN	Fz	Neuroscan	5.34 (0.86) [3.67–7.02]	
			Flex	4.82 (0.78) [3.31–6.33]	
	P300	Pz	Neuroscan	10.2 (1.73) [6.85–13.6]	
			Flex	5.57 (0.75) [4.11–7.04]	
Pre-registered	N170-face upright	P7	Neuroscan	7.21 (1.10) [5.05–9.36]	
			Flex	6.90 (0.88) [5.19–8.62]	
		P8	Neuroscan	11.2 (1.64) [8.01–14.5]	
			Flex	8.44 (0.97) [6.55–10.3]	
	N170-face inverted	P7	Neuroscan	7.80 (0.95) [5.94–9.66]	
			Flex	6.33 (0.72) [4.92–7.74]	
		P8	Neuroscan	11.5 (1.42) [8.68–14.3]	
			Flex	10.5 (1.25) [8.02–12.9]	
	N170-watch upright	P7	Neuroscan	7.67 (1.80) [4.14–11.2]	
			Flex	5.26 (1.16) [3.00–7.53]	
		P8	Neuroscan	8.63 (1.54) [5.62–11.6]	
			Flex	7.30 (1.70) [3.97–10.60]	
	N170-watch inverted	P7	Neuroscan	6.04 (0.67) [4.74–7.35]	
			Flex	6.61 (1.31) [4.05–9.17]	
		P8	Neuroscan	7.82 (1.47) [4.95–10.7]	
			Flex	5.07 (0.97) [3.18–6.96]	

P300 results

Signal quality

For the P300 paradigm, there was very strong evidence that Neuroscan SNR was larger than Flex SNR (6.60 vs. 5.10; Table 3). In addition, there was a statistically greater number of Flex epochs rejected. These results suggest that, overall, the signal quality of Neuroscan was better than Flex in the P300 paradigm.

Waveform similarity

The P300 waveforms were the responses to deviant tones in the active listening condition. Figures 3D and 3H show P300 waveforms for Fz and Pz, respectively. The waveforms of both systems were very similar (ICCs > 0.93). They did not include zero in their 95% confidence intervals and so were considered significant.

The P300 peak

Figure 3H contrasts group-average P300 ERP waveforms measured at Pz. We conducted a Bayesian t-test between P300 peak values of each system. Results indicated substantial evidence that there was no difference between the P300s captured by the two systems, BF10 = 0.26.

For consistency between P300 and MMN analyses, we calculated a post-hoc time window by calculating the maximum value of the overall grand average ERP between 200 and 400 ms. We centred a new time window of 100 ms on the timepoint at which this maximum value occurred. Our new time window was calculated as 198–298 ms. After calculating P300 values for each participant based on this window, we conducted another Bayesian t-test between the systems. These results mirrored our initial analysis and suggested that there was no difference between the P300 peaks measured by the two systems, BF10 = 0.32. See Table 4 for descriptive statistics.

P300 SNR values for each system are presented in Table 5. Values reflecting both the pre-registered and post-hoc time windows are shown. Figure 5 displays the topographic distributions of P300 peaks at Pz for Neuroscan (C) and Flex (D). As with the MMN, we observed a moderate correlation of channel values between systems (r = 0.60, p < 0.001).

N170 results

Signal quality

For the N170, there was substantial evidence that the Neuroscan SNR was larger than Flex (6.04 vs. 5.12; Table 3). However, there was no difference in the number of rejected epochs between the two systems.

Waveform similarity

Figure 6 contrasts Neuroscan and Flex N170 ERP waveforms over the left (P7) and right (P8) hemispheres for each stimulus type with ICCs found in their respective panels. As none of the 95% confidence intervals included zero, we considered all ICCs to be significant. Right hemisphere waveforms (P8) exhibited a higher degree of similarity than left hemisphere waveforms (P7).

Figure 6 Group N170 ERP waveforms and intraclass correlations [95% confidence interval] for Neuroscan and Flex systems by stimulus type.

(A, C, E, and G) depict left hemisphere waveforms (P7). (B, D, F, and H) depict right hemisphere waveforms (P8). Intraclass correlations were considered significant if the 95% confidence interval did not include zero.

The N170 peak

We performed a Bayesian t-test between the N170 peaks in the upright face condition at channel P8 (Fig. 6B). Results indicated very strong evidence that the N170 peaks measured by the two systems were different, BF10 = 35.88. We also performed a Bayesian t-test between the N170 peaks in the upright face condition at channel P7 (Fig. 6A). Results also indicated strong evidence that the N170 peaks measured by the two systems were different, BF10 = 22.59. Table 6 for descriptive statistics.

Table 6 N170 peak and latency group average descriptive statistics for left hemisphere (P7) and right hemisphere (P8). Figures represent mean values [95% confidence interval].

N170 amplitude measures were calculated as the average amplitude between 120 and 220 ms. Latency measures were calculated as the time point corresponding to the minimum amplitude between 120 and 220 ms.

Site	Stimulus Type	Measure	System	Upright	Inverted	
P7	Faces	Amplitude	Neuroscan	−1.51 [−2.40 to −0.61]	−2.84 [−3.79 to −1.89]	
			Flex	−2.82 [−3.93 to −1.70]	−3.69 [−5.02 to −2.37]	
		Latency	Neuroscan	182 [177–188]	191 [185–196]	
			Flex	181 [172–190]	188 [178–198]	
	Watches	Amplitude	Neuroscan	0.58 [−0.38 to 1.55]	0.20 [−0.83 to 1.24]	
			Flex	−1.38 [−2.55 to −0.22]	−1.57 [−2.83 to −0.31]	
		Latency	Neuroscan	178 [166–189]	178 [167–190]	
			Flex	176 [162–189]	182 [170–194]	
P8	Faces	Amplitude	Neuroscan	−2.37 [−3.78 to −0.95]	−3.43 [−4.77 to −2.09]	
			Flex	−3.93 [−5.65 to −2.21]	−4.56 [−6.30 to −2.81]	
		Latency	Neuroscan	181 [176–186]	188 [183–193]	
			Flex	178 [173–184]	184 [179–190]	
	Watches	Amplitude	Neuroscan	1.40[−0.17 to 2.97]	1.43 [−0.25 to 3.10]	
			Flex	−0.54 [−2.32 to 1.24]	−0.16 [−1.84 to 1.51]	
		Latency	Neuroscan	191 [186–197]	190 [184–196]	
			Flex	191 [183–198]	192 [184–200]	

Signal-to-noise ratio values for each of the N170 stimulus conditions at P7 and P8 are displayed in Table 5 and Fig. 7 displays the topographic distributions of N170 peaks across the stimulus conditions. Again, topographies of the two systems were moderately correlated (r = 0.56, p < 0.001).

Figure 7 Topographic distribution of N170 ERP signals for each of the stimulus conditions.

Values represent the average signal between 162 and 200 ms for Neuroscan (A, C, E, and G) and Flex (B, D, F, and H).

SSVEP results

Figure 8 shows the SNR values for each system across the frequency spectrum (1–30 Hz). The Bayesian t-test at the stimulus frequency provided decisive evidence that the magnitudes of the system SNRs differed, B10 = 51764. The SNR z-scores for both systems exceeded 1.96 at the stimulus frequency (Table 7). This indicated that, although the Neuroscan SSVEP response was larger than the Flex response, both systems were still capable of registering an SSVEP signature.

Figure 8 Signal-to-noise ratio (SNR) across frequency spectrum, 1–30 Hz.

(A) Neuroscan SNR. (B) Flex SNR. SNR values were averaged across occipital electrodes (O1, Oz, O2) for each system.

Table 7 Signal-to-noise ratio (SNR) descriptive and inferential statistics.

SNR values represent average of occipital sites (O1, OZ, O2). The values were considered significant if z-scores were greater than 1.96 (p < 0.05).

System	Mean (SD)	z-score	
Neuroscan	19.94 (9.51)	12.47	
Flex	8.98 (4.80)	11.22	

Resting state results

Figure 9 depicts the topographical alpha power (8–12 Hz) distribution for eyes open and eyes closed condition for each system. Table 8 provides descriptive statistics for alpha power in each condition. Figure 10 contrasts power for each system and condition across the 1–30 Hz frequency spectrum. The Bayesian ANOVA indicated very strong evidence for a main effect of condition (B10 = 32.27) and substantial evidence for no effect of system (B10 = 0.28). Thus, both Neuroscan and Flex measured similar differences in alpha power between eyes open and eyes closed conditions.

Figure 9 Resting state alpha band (8–12 Hz) scalp topography maps.

(A) Neuroscan ‘eyes open’. (B) Neuroscan ‘eyes closed’. (C) Flex ‘eyes open’. (D) Flex ‘eyes closed’.

Table 8 Resting state alpha power descriptive statistics for each condition and system.

Power was measured as the power averaged over occipital sites (O1, Oz, O2) across frequencies 8–12 Hz.

System	Measure	Mean	Standard error	95% CI	
Neuroscan	Eyes Open	0.01	0.00	[0.01–0.02]	
	Eyes Closed	0.05	0.01	[0.02–0.07]	
Flex	Eyes Open	0.01	0.00	[0.01–0.02]	
	Eyes Closed	0.07	0.02	[0.02–0.11]	

Figure 10 Power spectrum in ‘eyes open’ and ‘eyes closed’ conditions across frequencies (1–30 Hz).

(A) Neuroscan power. (B) Flex power.

Discussion

The aim of the current study was to validate a consumer-grade EEG system, Emotiv EPOC Flex. To achieve this, we simultaneously recorded EEG with Flex and Neuroscan across an array of tasks designed to provide comparisons in temporal and phasic domains. In the time domain we measured auditory and visual ERPs and in the frequency domain we measured SNR response to a flickering screen and alpha power during a resting state task.

Our first main finding was that Flex recorded auditory ERPs similar to those measured by Neuroscan. This was evident in the finding that the magnitude of MMN and P300 peaks were equivalent between systems. Further, ERP waveforms in both the passive and active listening conditions were very similar between systems as were the scalp topographies. The similarity of signal distribution across all electrodes is particularly noteworthy as it suggests that even sites that were distant from the main sources of the ERP-relevant signals still show similar measurements between the two systems. Comparison of the face-sensitive N170 waveforms captured by the two systems also showed a high degree of similarity, particularly over the right hemisphere. Again, scalp topographies demonstrated a high degree of signal similarity, even at sites distant from occipito-temporal areas. Although we observed similarity between face-related signals we did find substantial evidence that the actual magnitudes of the N170 peaks were different between systems. While this finding indicated that the N170 amplitudes measured by Flex and Neuroscan were different, it does not mean that one of the systems was incapable of measuring an N170 ERP. In fact, Flex actually measured a larger-amplitude N170 peak. This was likely due to the constraints of a simultaneous setup wherein system sensors could not be situated in the exact same locations. As it was the system being validated, we placed Flex sensors in the precise location of interest (P8 in this case) and Neuroscan sensors immediately adjacent along an axis towards Cz. This likely positioned Flex in an advantageous position for capturing face-related neural signals and contributed to the larger N170 peak at P8. Indeed, a recent study examining localisation of face-related brain signatures found the largest N170 effect at PO10 (Gao et al., 2019). Thus, in the current study, the Flex sensor was in closer proximity to PO10 than the Neuroscan sensor. A similar pattern and explanation has been noted in an N170 validation of another Emotiv system (Emotiv EPOC; De Lissa et al., 2015).

Another potential explanation for the larger amplitude N170 peak observed for Flex may be the fact that it exhibited a smaller P1 value relative to Neuroscan (see Fig. 6). This highlights the sensitivity of these types of comparisons to the calculation method with which ERP component and peak values are derived. A ‘fairer’ comparison may have been a so-called peak-to-trough comparison in which we calculated N170 values based on the difference between P1 and N1 values. Irrespective of the calculation method, it is the implicit aim of validation studies such as the current one to determine whether the system being validated collects accurate data, not whether it collects data identical to that of another system. Considering the high degree of morphological similarity between N170 waveforms, it is apparent that Flex is indeed capable of indexing visual ERPs.

Our findings also demonstrate that Flex measures acceptable spectral EEG data. This is evidenced by results showing higher alpha power during eyes closed relative to eyes open conditions. Further, we observed no differences in alpha power between Flex and Neuroscan. We did, however, observe a large difference between the systems in the SNR measured in an SSVEP task. The Neuroscan SNR was orders of magnitude larger than Flex SNR. Why this would be the case is not immediately evident. However, we speculate that the lower recording resolution of Flex—129 Hz vs. 1000 Hz for Neuroscan—may have introduced temporal smearing wherein in the EEG response at the stimulus frequency was binned to adjacent frequencies thereby lowering SNR values. Indeed, even minute variations in system timing can degrade evoked EEG signatures (Hairston, 2012; Ouyang, Sommer & Zhou, 2016) and, given their often-high frequency stimulation rates, SSVEPs are particularly susceptible. Though Flex measured smaller SNR values compared to Neuroscan, the SNR peaks observed at the stimulus frequency were nonetheless significant compared to values at surrounding frequencies.

Another finding was that overall signal quality was lower for Flex than for Neuroscan in the P300 task. While we did not have enough evidence for SNR differences in the MMN task to make a conclusion, the number of threshold-rejected epochs in each of the auditory tasks was higher for Flex than for Neuroscan. For the N170, SNR was greater for Neuroscan than Flex, although the number of rejected epochs was not statistically different. Overall, the differences in SNR were not surprising as Neuroscan has a much higher resolution than Flex (24-bit vs. 14-bit, respectively) and would thus be expected to be more sensitive to signal fluctuation. This would be particularly true when calculating SNR over the entire scalp where many of the measurements are distant from the signal generator of interest and thus more susceptible to noise-distortion. The reason for disparity between the auditory and usual modalities was likely due to the fact that participants were given explicit opportunity to blink their eyes (during ‘blink’ screens) while performing the N170 task, which would have placed most motion artefacts outside of epoch limits. Explicit opportunities to move or blink were not present during the auditory tasks. As such, there was likely more motion artefacts present in these epochs and Flex was more sensitive to these. However, considering that the median number of rejected artefacts represented approximately 1% of trials in both passive and active conditions, we did not consider signal quality to be an issue with Flex.

One limitation of this study is the nature of the simultaneous setup of the two EEG systems. Because we wished to test whether the signals measured by the two systems were similar, we placed the respective electrodes immediately adjacent to each other at each site. While this allowed us to assess the similarity of signals, it also created a situation in which we could not guarantee that there was no low-impedance electrical bridging between the electrodes of the two systems. In other words, it was possible that the signals we measured with Flex were not the same as if Flex had been used alone. Unfortunately, to mitigate this limitation required either a between-subjects design (one system per participant) or to do two separate setups with each participant. The downside to the first solution is that there is a loss of statistical power to detect differences between the systems. The downside to the second solution is the potential for differences in intra-individual participant characteristics from system to system (e.g. fatigue, motivation). Therefore, we deemed the advantages of a dual setup greater than the disadvantages.

Another potential limitation is the fact that is not possible to quantify the exact impedance value of Flex. The Emotiv software uses a ‘traffic light’ impedance feedback system wherein users adjust electrodes until an indicator light turns green, which signifies impedance of less than 20 kΩ. This may concern some researchers as many protocols aim for impedances of less than 5 kΩ. However, the less-than-5 kΩ convention is typical of ‘passive’ EEG systems that use a ground circuit to subtract common mode voltage and achieve clean EEG recordings (Luck, 2014). Flex is an ‘active’ EEG system that uses a common mode sense and driven right leg to optimise EEG signal. Although explanation of the technical aspects of passive and active systems is outside the scope of this paper, we note that because of their circuitry, impedance values are less of a concern with active systems (Keil et al., 2014). Additionally, work by Kappenman & Luck (2010) suggests that while high-impedance setups may result in lower signal-to-noise ratio, this can be attenuated with higher trial counts, filtering, and artefact rejection.

In summary, these results support Flex as a valid device for EEG and ERP research. Naturally, there are compromises when choosing a lower-cost, consumer-grade EEG devices over more expensive research-grade devices. Both the lower bit rate and the lower sampling rate of Flex mean that it is not as sensitive as Neuroscan to small variations in EEG signals. This can also mean that these devices are ‘noisier’ than research-grade devices. However, these are shortcomings that are typical to all devices in this class and, for some researchers, are offset by their affordability and flexibility.

Conclusions

Overall, our findings suggest that Flex is capable of collecting research-quality data. It can register both event-related and spectral EEG signatures similar to those of a research-grade system. The validity of the Flex system makes it a suitable alternative to research-grade systems; an alternative that may be appealing to some researchers for whom traditional lab-based EEG systems are cost prohibitive. Additionally, the easy, convenient setup makes Flex suitable for use with atypical populations (e.g. children and sensitive individuals) by ameliorating participant discomfort experienced from long setup times and the use of electroconductive gel. Finally, the portability of Flex allows it to be used outside of the laboratory and in more naturalistic settings. By mitigating financial, methodological, and practical barriers, Flex may facilitate neuroscience research on topics and in locations in which it would otherwise not be feasible.

Supplemental Information

Supplemental Information 1 Emotiv EPOC Flex saline and Neuroscan dual-setup.

Click here for additional data file.

We would like to thank Geoff Mackellar, Bill King, and Mark Irwin of Emotiv who provided technical assistance with the Emotiv EEG devices. We would also like to thank Paul Sowman who provided support with the SSVEP analysis as well as Peter de Lissa who provided the stimulus for the N170 paradigm. Finally, we would like to thank the participants who volunteered their time.

Additional Information and Declarations

Competing Interests

Author Contributions

Human Ethics

Data Availability

Nikolas S. Williams is employed on a research fellowship that is funded by an industry partnership grant between Macquarie University and Emotiv. Genevieve McArthur and Nicholas Badcock are Academic Editors for PeerJ.

Nikolas S. Williams conceived and designed the experiments, performed the experiments, analysed the data, prepared figures and/or tables, authored or reviewed drafts of the paper, and approved the final draft.

Genevieve M. McArthur conceived and designed the experiments, authored or reviewed drafts of the paper, and approved the final draft.

Bianca de Wit conceived and designed the experiments, authored or reviewed drafts of the paper, and approved the final draft.

George Ibrahim performed the experiments, analysed the data, authored or reviewed drafts of the paper, and approved the final draft.

Nicholas A. Badcock conceived and designed the experiments, authored or reviewed drafts of the paper, and approved the final draft.

The following information was supplied relating to ethical approvals (i.e. approving body and any reference numbers):

This research was approved by the Macquarie University Human Research Ethics Committee (HREC; Ref. #5201831203493).

The following information was supplied regarding data availability:

Data and processing/analysis code are available at Open Science Framework: Williams NS, Badcock NA. 2020. A Validation of Emotiv Flex. OSF. Data and code. Available at https://osf.io/zj3f5/.

The ‘Raw Data’ is in the ‘Emotiv EPOC Flex Saline Validation’ folder which contains subfolders for each paradigm used in the study. These subfolders contain ‘behav’ and ‘eeg’ folders. ‘behav’ is the stimulus script output generated by MATLAB. Within the ‘eeg’ folders are raw EEG data files for both ‘emotiv’ and ‘neuroscan’.

Processing and analysis scripts are available in the ‘scripts’ folder. Procecssing scripts can be found within the ‘matlab_scripts’ subfolder. Analysis and visualisation scripts may be found within the ‘r_scripts’ folder.

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
