# Peer review of "A validation of Emotiv EPOC Flex saline for EEG and ERP research"

_PeerJ, doi:10.7717/peerj.9713_

## Round 0.1 · original submission · Major Revisions

Your manuscript has now been seen by 2 reviewers. You will see from their comments below that while they find your work of interest, some major points are raised. We are interested in the possibility of publishing your study but would like to consider your response to these concerns in the form of a revised manuscript before we make a final decision on publication. We, therefore, invite you to revise and resubmit your manuscript, taking into account the points raised. Please track all changes in the manuscript text file.

Reviewer 1 ·

Basic reporting

- The first sentence of the introduction is plain wrong. Note that action potentials cannot be the biophysical origin of the scalp-recorded EEG signal, as they do not sum up to open fields (in contrast to post-synaptic potentials or pyramidal neurons). Please consult well-established references in the field of EEG physics for details (e.g. the works of Paul Nunez).

- Repeatedly, statements in the manuscript imply ignorance of the inverse problem in EEG interpretation (direct inferences sensor to source cannot be made). Note that EEG was not measured „in the right hemisphere“ as stated in the abstract, but „over“ the right hemisphere. Indeed, the measurement of EEG asymmetries is already a highly complicated issue, and only the statement „over“ correctly expresses that signals measured over the right hemisphere may (or may not) originate in the right hemispheric regions of the brain. Again, the authors might benefit from consulting literature for details (e.g.: https://sapienlabs.org/the-inverse-problem-in-eeg/)

- The introduction ignores early work on signal quality of EMOTV systems. Specifically, Debener et al., (2012, Psychophysiology) presented the precursor of the EMOTIV Flex system, by combining the EMOTIV amp with easycap electrodes and reporting high-quality auditory oddball P300 ERPs even in outdoors free walking conditions (replication published by De Vos et al., 2014, Int J Psychophysiology). The same group also compared the EMOTIV amps with a research-grade amplifier in a P300 speller BCI paradigm and found very similar online performance (De Vos et al., 2014, J Neural Eng). Given that the Flex EMOTIV system is based - to a large extent – on the very same hardware as the EMOTIV epoc system, it is necessary to at least discuss those papers in the introduction. In fact, given that the most critical specs of the Flex and the epoc system are identical (14 bit, 128 Hz AD rate, etc), the introduction should also briefly summarize similarities and differences between Flex and epoc hardware.

Experimental design

- While the overall design is convincing, two issues captured my attention. First, can the authors really guarantee that bringing Flex and neuroscan electrodes next to each other did not generate low impedance electrical bridges? Otherwise the limitations resulting from such electrical shortcuts should be discussed (i.e. is cannot be guaranteed that the Flex system would generate the same signal quality when used alone). Please provide at least a close-up picture of the electrode arrangement and discuss this limitation. Another issue is the use of different filter settings. Why was the neuroscan data recorded with a 1 Hz high-pass filter? It would have been much better to use a filter matching the EMOTIV filter settings (at least offline). This issue requires some discussion as well, as a 1Hz filter clearly alters the P300 morphology (i.e. attenuates P3 amplitudes).
- Using the number of remaining trials as a measure of EEG quality is not convincing, in particular when a rather dull amplitude-based rejection criterion was used. Note that the amplitude of EEG signals depends on many factors (for instance, the distance to the reference location, which will be larger for a person with a larger head even if identical 10-20 locations were used) and does not separate signal from artefact very well (e.g. some individuals have eye blinks below 120 uV, others have alpha oscillations above 150 uV). A much better criterion for ERP comparison is the signal-to-noise-ratio (signal amplitude divided by the rms of the pre-stimulus signal, for instance). Please replace your trial-number results with a proper report of SNR values.

Validity of the findings

- While the ERP morphology comparison at single channels is well justified, this analysis missed another key feature of EEG signal quality, namely the topographies. Unfortunately, topographies are only provided for some analyses. Please add topographic illustrations and a statistical comparison of the topographies. Very similar topographies would support your claim that the Flex system can collect research-grade signal quality.

- the acknowledgments state that the paper was supported by the company EMOTIV. The authors should add a statement on potential conflicts of interest.

Additional comments

This is an interesting paper comparing ERPs, SSVEPs and resting-state EEG spectra that were recorded concurrently with a new consumer-grade EEG system and a research-grade system. Overall, the results reveal very similar signals, which led the authors to conclude that the consumer-grade system allows recording research-grade EEG signals. This finding is not really surprising, given previous reports on the issue.

There is much to like about this paper, which is well written and easy to follow. Moreover, the experimental design is straight-forward and the Bayesian approach used for the statistical analyses makes perfect sense, as it allows interpreting similarities and differences. However, the following issues require consideration:

Reviewer 2 ·

Basic reporting

The manuscript is written well and presents data clearly within the presented tables and figures. The authors have also provided a good literature review of EEG technology and the emergence of low-cost EEG alternatives. I feel the rationale around the choice to validate the Flex Emotiv EEG could be communicated more strongly in the introduction. For example, the importance of why the Flex model has been chosen for validation has not been communicated other than “providing greater sensor coverage” (line 156) compared to three other brands of portable EEG device. If the specifications of the Flex EEG device were stated within the introduction (e.g., sampling rate, number of electrodes, method for reducing and measuring impedances, details of the data acquisition software used, etc) it would help establish clearly the pros and cons of this system relative to a lab-based EEG set-up, and hence, the importance for the present validation study.

Some other points are included below for your reference:

-Line 133 – Could the authors specify which “involved set-up procedures” and “lab-based limitations” are being alluded to regarding lab-based EEG, and describe how Flex may overcomes these problems?

-Line 288 – I appreciate the transparency of reporting the sampling rate of your specific Emotiv Flex device as being 129.05 Hz. The accuracy of an EEG systems’ sampling rate is highly important when suggesting an EEG device to be research-quality, so feel this should be considered in more depth in the body of the manuscript rather than in a footnote. It might be helpful to discuss what variability in sampling rates one might expect from low-cost EEG devices, and what impact (negligible or not) that such variations in hardware may have on research accuracy. Further, it would be highly beneficial to describe your method for determining the precise sampling rate of your Flex unit to allow other researchers to do the same to their own devices.

-It would be fruitful to discuss the trade-off of using low-cost EEG devices compared to traditional, lab-based equipment for research. For example, the added flexibility of the Flex system is off-set by a lowered sampling rate of 128 Hz compared to the much higher sampling possible from research-grade EEG. A more balanced assessment of the strengths and weaknesses of the Flex EEG device should be offered in the discussion to concede some of the current limitations of portable EEG technology for research.

-Line 283. Please provide more comment regarding the Flex’s impedance measurement system. What does a connection of “green” in the Flex acquisition software equate to in kΩs?

-There are inconsistencies regarding specifics of the method used to different analyses and the subsequent data presented in the results. For example, Lines 362 and 363 imply assessing the N170 only at P8, but data for both P7 and P8 electrodes is presented in Figure 5 and discussed later in the manuscript. Please check for consistency between these details explained in the method and subsequently reported output.

-Lines 171, 302, 309 and 315 – The citations/ URL’s provided in text in these lines of text appear to be broken. Please revise.

-Line 509 – remove the second “is” in the sentence “it is the implicit aim of validation studies such as the current one is to determine …”

-Line 276 – recommend “ml” be written as “mL”

-Line 113 – remove period at the end of citation “(…2015 for a review). and represent a ”...

Experimental design

The experimental design used to produce the desired ERP components and EEG power spectra were sensible. I feel this is a valuable research area and the manuscript is working towards addressing a relevant and meaningful research question. I have no major comments as to the methodology used, but have included some suggestions regarding the reporting of the methodology to improve the ability to replicate.

-Lines 318, 328, 331 and 333 – Please provide explicit description of the settings used within the Fieldtrip functions described at these lines. For example, line 318 should describe details of the hanning window used, including the length, step and overlap of the window used to allow precise replication of this FFT process in future.

-Line 231 – Please describe in what location relative to the Flex electrodes the Neuroscan electrodes were placed adjacent to each other on the scalp. For example, the direction and approximate distance of Neuroscan adjacent to Flex electrode would be valuable for replication.

-Line 191 to Line 200 – The number of blocks, number of tones-per-block, and overall MMN task length, should be reported in this section for replication of the MMN paradigm.

-Line 213 – Wording appears to suggest the N170 delivered in a single block of 300 trials, with a self-timed break after 150 trials. Can the authors please confirm/ elaborate upon the administration of the N170 task? For example, what was the task length for the N170 paradigm used?

-Line 225 – Please describe what materials were used in the SSVEP paradigm. For example, was the same monitor placed at 50cm away as in the N170 paradigm used here?

-Line 226 – Duration of the eyes-open RS task should be stated. Were any instructions provided to participants relating to minimising eye-blinks and movements during this task? Furthermore, procedure for the ‘eyes-closed’ RS paradigm was not described at all. Please confirm details of this task, including the duration of the eyes-closed block, and specifying any instructions provided to participants.

-Please provide citation/s supporting the methods used for calculating the SNR as described in lines 320 to 322.

Validity of the findings

It is unclear why the researchers recorded EEG from 16 scalp electrode sites, but only report comparisons of single electrodes or small groups electrodes for most of their experimental paradigms. A more comprehensive manner to demonstrate Flex being a robust EEG device would demonstrate comparable signal quality to Neuroscan at all the selected International 10:20 electrodes sites. For example, the ICCs achieved during the N170 paradigm were lower at P7 compared to position P8, indicating that different electrodes may have different recording quality relative to the Neuroscan. It may be that the further away an electrode is placed from central scalp locations, or locations where ERP components are reliably detected, that the SNR of the Flex diverges significantly from a research-grade device like Neuroscan. Given the amount of data acquired across 16 electrode sites, I feel the authors have the opportunity to provide more comprehensive analysis of the EEG across all the different International 10:20 electrodes recorded from to conclusively show Flex and Neuroscan do not differ across any of the chosen electrode locations used.

Additional comments

Thank you for the opportunity to review this manuscript. The authors present a thorough, well-devised experiment to validate a low-cost EEG device detecting both ERP and temporal EEG date. This is a valuable research area, and the breadth of data collected across several common EEG paradigms is an intelligent way to compare the Emotiv Flex EEG device’s signal quality against a research-grade Neuroscan EEG device. The manuscript provides a good overview of the relevant literature supporting the five paradigms used but is less clear in establishing the advantages that the Emotiv Flex EEG device offers over traditional EEG and other low-cost EEG devices. This leaves the importance of validating this specific EEG device somewhat in question. For example, the authors could elaborate as to the technical specifications of the Flex EEG system in comparison to the other portable EEG systems (e.g., Neurosky, Muse 2 and EPOC) including the number of recording electrodes, sampling rate, data acquisition hardware and software), and methods of data acquisition (e.g., use of saline compared to electroconductive gel, methods for reducing electrode impedances, method for and reliability of recording electrode impedances) to demonstrate Flex as a credible research-grade EEG system.

The paradigms chosen to illicit a MMN, P300, N170 and the detection of alpha power and SNR ratios across 1 to 30 Hz power-bands were intelligent and demonstrate the Flex EEG system’s capabilities across several measures of EEG. However, I am unclear why such a small subsection of all the recorded EEG data acquired from these paradigms were presented within the results of this paper. It appears that limited/ no analysis was conducted on data collected from many of the electrode sites recorded from (i.e., Fp1, Fp2, F3, F4, FT7, FT8, CP3 and CP4) in preference of comparing signal quality at sites known to robustly show an ERP component of interest (e.g., Fz to assess MMN detection). I feel a more comprehensive demonstration that the Flex EEG system has comparable signal quality to the Neuroscan system at all 16 recording sites across numerous paradigms in order to be genuinely considered a potential research-grade EEG device. Additionally, some aspects of the procedure and specific aspects of the EEG data processing should be stated more explicitly to allow independent replication of findings.

Overall, further methodological explanation and statistical evidence is needed to convincingly meets the stated aim of the manuscript in showing the the Emotiv Flex is equitable to Neuroscan for research-standard EEG.

---

## Round 0.2 · Minor Revisions

Your manuscript has now been seen by the reviewer. You will see from the comments below that a number of constructive points are worth considering. We therefore invite you to revise and resubmit your manuscript, taking into account these points. Please highlight all changes in the manuscript text file and provide a point-by-point response letter.

Reviewer 2 ·

Basic reporting

• Line 69: the allusion to “different questions using different techniques” is vague. Could examples of such questions be explicitly/ immediately stated, or else a stronger segue to EEG oscillations provided?
• Lines 73 and 74: it is unclear why the Delta power is described here given you do not explore deep-sleep or detection of the delta band specifically in any analyses.
• Line 75: “myriad processes”
• Line 114: Replace comma with period after “(Eimer, 2012),”
• Line 151: Please describe some of the ways these systems have increased portability and easier set-up than lab-based devices.
• Line 154: Could you add a comment regarding in what capacity they have attracted the attention of researchers? Are you referring to they attract researchers due to their advantages over lab-based EEG, they are attracting researchers attempting to validate the systems, or both? That is, are low-cost EEG devices being adopted by researchers for experimental studies at present?
• Line 174/ Line 175: Was this a ‘Muse’ or ‘Muse2’ model as per your earlier listing in Line 152?
• Line 176: Given your earlier mention of ‘NeuroSky MindWave’ in Line 153, please provide an example (if available) of a validation of this system or else discuss the weighting of the different validations of these systems. That is, since the literature review provided shows that most prior interest has been towards validating Emotiv EPOC systems please comment as to how well validated the different low-cost EEG devices you’ve discussed currently are.
• Line 196: As stated above, the advantage of the “configurability” of the system is overstated given it has been possible to modify these systems to be configurable. Please comment/ revise.
• Lines 354 to 364: un-bold text in this section
• Line 536: “neuroscan” misspelled (missing 'o')
• Line 556: “Figure 3” on wrong line
• Line 798/799: un-bolden DOI
Note: I had difficulty aligning some figures with the relevant output being discussed in text. For example, I have missed the distinction between what the top (A, B, C, D) and bottom (E, F, G, H) panels of figure 3 are displaying. For example, it was unclear what the comparison between Panels D and H of figure 3 (as made on line 552) was intended to contrast. Are these deviant signals under the active listening from different 10:20 scalp locations? In addition, panels C and D of figure 5 are not clearly labelled in text (likely an error in the description in line 572). Please double-check all in-text descriptions and figures for clarity and ease of comprehension.

Experimental design

• As per previous review, the study is conducted to a high empirical standard.

The gap in the literature this validation addresses is more clearly outlined in the introduction, but the advantage of the Flex system as being “configurable” is overstated as other low-cost EEG devices can be modified to be configurable to be used with an EasyCap (see above comment). I also feel the distinction between the Flex model of the Emotiv and prior EPOC+ systems could be discussed further, with the emphasis of the types of analyses possible with a 32-electrode array over the 14-electrode array of other devices being the main advantage of the system. Hence, to show the benefits of this system above others, this unique aspect of the system need to be emphasized and clearly linked to the unique research questions that could be better answered using Flex > EPOC+ versions of Emotiv

Validity of the findings

Line 182: While the rigidity of the sensor placements in base EPOC is true, several of the prior studies you have cited (Debner et al 2012, Debner et al, 2014; Barham et al., 2017) have already validated modified versions of the EPOC in which the ridged sensory bracket was replaced with shielded EEG wires allowing a configurable EEG array. Hence, this identified limitation of low-cost EEG systems is not completely accurate. Please revise.

I feel the authors have demonstrated comparable signal quality between EEG systems across several EEG and ERP protocols, an important limitation of the system that needs to be acknowledged more directly is the colour-coded impedance measurement system. I appreciate the authors personal correspondence finding that a “green” equates to an impedance of “<20 kOhms” (line 341) but the implications of this is not considered beyond this point. To me, the connotations of the conclusion that Flex is “a suitable alternative to research-grade EEG system” means both that is able to acquire accurate EEG and ERP data (demonstrated well), but also that it is possible to publish data acquired with a system as per the typical reporting standards of a field. Given EEG research studies typically aim for (and report) achieving electrodes impedances < 5 (or sometimes < 10) kOhms, I feel the ambiguity both in the actual impedance that was achieved, and the much higher minimum impedance of the Flex cannot be ignored. While not an irreconcilable problem, it is a genuine drawback of the Flex that is not considered at all in the manuscript. I would appreciate the authors comments on this limitation and a slightly more balanced consideration of the trade-off of this low-cost system has over lab-based devices.

Additional comments

This is a well written manuscript with good methodological rigour. I appreciate the inclusion of the comparison Table of the different low-cost EEG systems in the introduction, additional explanation of the methodologies of the different tasks, and additional analyses regarding the SNRs of the systems. Aspects of the basic reporting need to be reviewed (see above suggestions). Regarding the “gap” in the literature, I would encourage the authors to more clearly differentiate the advantages of this Flex model of and Emotiv EEG system from the EPOC+ (standard and modified versions) which have undergone several past validations in the past. Given the better sensor array at the main strength of this system over the 14-channel EPOC+ studies, discussion of the types of studies which can be achieved with 32-electrode arrays that cannot be achieved with 14-electrodes would be a useful. In addition, the “configurability” of the system is overstated given several prior studies which have shown modifications are available to the EPOC+ to allow it to be used with an EasyCap at configurable 10:20 locations. Finally, the Flex does provide appear to have potential as a research-grade EEG system, but some limitations of the system may be overlooked/ downplayed to make the conclusion about the system appear stronger than they potentially are. For example, the subtle, but important limitation of the Flex colour coded “traffic-light” impedance measuring method should be considered as part of the shortcomings of the device and final conclusion about the value of the system.

---

## Round 0.3 · accepted · Accept

Thank you for the detailed response letter. We are delighted to accept your manuscript for publication.